# Postoperative Findings of Common Foot and Ankle Surgeries: An Imaging Review

**DOI:** 10.3390/diagnostics12051090

**Published:** 2022-04-27

**Authors:** Maryam Soltanolkotabi, Chris Mallory, Hailey Allen, Brian Y. Chan, Megan K. Mills, Richard L. Leake

**Affiliations:** Department of Radiology & Imaging Sciences, University of Utah School of Medicine, Salt Lake City, UT 84132, USA; chris.mallory@hsc.utah.edu (C.M.); hailey.allen@hsc.utah.edu (H.A.); brian.chan@hsc.utah.edu (B.Y.C.); megan.mills@hsc.utah.edu (M.K.M.); richard.leake@hsc.utah.edu (R.L.L.)

**Keywords:** postoperative foot and ankle, foot and ankle imaging, osteotomy, arthrodesis, articular implants, post-traumatic fixation

## Abstract

Foot and ankle surgery is increasingly prevalent. Knowledge of the mechanisms underlying common foot and ankle deformities is useful in understanding surgical procedures used to restore normal biomechanics. As surgical techniques evolve, it is important for the radiologist to be familiar with these procedures, their expected postoperative appearance, and potential complications. This article reviews the key imaging findings of a variety of common and important foot and ankle surgical procedures.

## 1. Introduction

Foot and ankle surgery is increasingly prevalent. In 2000, a total of 548,214 foot and ankle surgeries occurred in the Medicare population. An approximate direct economic burden of USD 11 billion was attributed to these procedures [1]. Additionally, osteoarthritis is a common reason for primary care visits and surgical consultation. The quantity and variety of foot and ankle surgeries has increased over time. Familiarity with these procedures, their expected postoperative imaging findings, and common complications are important for the general radiologist as well as the musculoskeletal subspecialist.

## 2. Osteotomies

### 2.1. Hallux Valgus and Metatarsus Primus Varus

Hallux valgus (HV), defined as abnormal fixed abduction of the first metatarsophalangeal (MTP) joint, is the most common form of forefoot malalignment. It is often seen concurrently with metatarsus primus varus (MPV) or abnormal adduction of the first metatarsal related to the other metatarsals. Both HV and MPV can cause foot pain, stiffness, and chronic irritation of the overlying skin.

Foot radiographs are the mainstay for preoperative imaging of HV and MPV. As most foot and ankle deformities have a dynamic biomechanical component, non-weight-bearing radiographs tend to underestimate deformity severity.

Many radiographic measurements have been described to define HV and MPV. Three of the most commonly used measurements include the hallux abductus angle (HAA), the first intermetatarsal angle (IMA), and the metatarsal sesamoid position (MSP) (Figure 1A,B). Normal HAA and IMA are less than 15 and 10 degrees, respectively [2]. The MSP describes the relationship between the longitudinal axis of the first metatarsal and the position of the tibial sesamoid. This is measured on a 7-point scale, with positions 1–3 depicted as normal [3].

As the abduction deformity progresses, the first metatarsal head slides medially in relation to the sesamoids, the medial bony prominence (bunion) increases, and the medial supporting structures attenuate. Many patients do not respond to conservative management; subsequently, multiple surgical techniques are used to address the underlying malalignment.

The choice of procedure depends on the severity of the deformity. More severe deformities generally require more proximal metatarsal osteotomy or first tarsometatarsal (TMT) arthrodesis. Figure 2 depicts an overview of various techniques used for correction of HV/MPP.

### 2.2. Akin Osteotomy

The Akin procedure is a medial-wedge osteotomy of the first proximal phalangeal base, typically performed in combination with a first metatarsal osteotomy procedure (Figure 3A,B) [4]. A single screw or cerclage wire are typically used for osteotomy fixation.

### 2.3. Chevron Osteotomy

Chevron osteotomy, a V-shaped osteotomy of the first-metatarsal head performed in the medial-to-lateral direction, is best for mild deformities [5,6]. Traditionally, the dorsal and plantar osteotomy limbs were equal in size, but more recently, the dorsal limb has been cut longer to accommodate the fixation screw [7]. The distal fragment is then translated laterally and typically fixed with one or two bicortical screws (Figure 3A). Postoperatively, lateral radiographs are important to ensure the screws do not violate the metatarsal–sesamoid articulation and do not protrude into adjacent soft-tissue structures.

### 2.4. Reverdin Osteotomy

The Reverdin osteotomy consists of medial rotation of the articular surface of the first metatarsal head [8]. Radiographs lack sensitivity to depict the exact position of the cartilage-covered metatarsal head. Thus, most surgeons rely on intraoperative findings when making the decision to perform a Reverdin osteotomy.

### 2.5. Scarf Osteotomy

A longitudinal osteotomy on the first-metatarsal shaft is performed from medial to lateral, often for moderate degrees of hallux valgus [9]. The distal fragment is translated laterally and fixed with two screws (Figure 3B). Lateral postoperative radiographs are especially useful in depiction of screw orientation.

### 2.6. First-Metatarsal Base Closing Wedge Osteotomy

For severe HV/MPV deformities, procedures at the base of the first metatarsal may be required. One such procedure is the first-metatarsal base-closing-wedge osteotomy, which is a lateral oblique osteotomy fixed with at least one bicortical screw [10].

### 2.7. Lapidus Arthrodesis

Lapidus arthrodesis is another option for severe deformities, which combines a first-tarsometatarsal arthrodesis and closing-wedge osteotomy of the first-metatarsal base (Figure 3C) [11]. A potential complication is excursion of the screws into the naviculocuneiform joint; this is best appreciated on lateral radiographs.

### 2.8. First Metatarsophalangeal Arthrodesis

Arthrodesis is an option for patients with severe HV/MPV deformities and first MTP osteoarthritis (Figure 3D,E). Joint instability, as can be seen in patients with rheumatoid arthritis, is another indication of arthrodesis. The goal postoperative alignment is approximately 10 degrees of phalangeal abduction (10-degree hallux abductus angle), 10 degrees of dorsiflexion, and 0–5 degrees of valgus.

### 2.9. Bunionectomy and Bunionettectomy

Resection of the medial and lateral eminences of the condyles of the first and fifth metatarsals—termed bunionectomy and bunionettectomy, respectively—are very common adjunct procedures in the treatment of HV/MPV. They are rarely performed in isolation, as they do not contribute to realignment of the bones. Postoperative radiographic evaluation is mainly performed to assess for complications such as infection or bunion recurrence.

### 2.10. Second-Metatarsal Shortening

Second-metatarsal shortening is often performed for correction of long-second-toe syndrome and metatarsalgia that has failed conservative treatment. In long-second-toe syndrome, the second toe is longer than the hallux and third toes, resulting in hammertoe deformity, callus formation, ungual lesions, and pain [12]. Surgical options include joint-preserving metatarsal-shortening osteotomy (Weil osteotomy), resection arthroplasty, and arthrodesis [13]. An osteotomy is made from the dorsal aspect of the metatarsal neck, parallel to the plantar surface of the foot. The distal fragment is then translated proximally to achieve the desired shortening (Figure 4A,B). Decreased dislocation rate, pain reduction, and resolution of soft-tissue callus are advantages of the Weil osteotomy. Postoperative improvement in range of motion does not always occur, and postoperative extension contractures and joint stiffness have been reported [14].

### 2.11. Calcaneal Osteotomy

The calcaneus plays an important role in alignment of the foot during weight-bearing. Calcaneal osteotomies are extra-articular, joint-preserving procedures utilized in the correction of pes planovalgus and pes cavovarus. Not only do calcaneal osteotomies correct alignment through repositioning of the calcaneal tuberosity, but they also change the direction of the pull of the Achilles tendon, transforming it into a corrective force. A variety of calcaneal osteotomies exist and can be broadly divided into translational osteotomies (medializing or lateralizing), closing-wedge osteotomies (Dwyer/modified Dwyer) [15], and rotational osteotomies (Evans or Z-osteotomy) [16]. A major contraindication to these procedures is subtalar osteoarthritis. In the presence of subtalar osteoarthritis, subtalar arthrodesis is preferred.

### 2.12. Single-Plane Translational Osteotomy

Single-plane translational osteotomies are simple transverse osteotomies of the posterior calcaneus. The calcaneal tuberosity may be translated medially, laterally, anteriorly, or posteriorly depending on the type of realignment required.

Medializing calcaneal osteotomy is usually performed for correction of pes planovalgus caused by posterior tibialis tendon dysfunction. It reduces strain on the deltoid ligament and other medial support structures. The osteotomy is usually fixed with two retrograde screws or a lateral plate and screw construct. A lateralizing calcaneal osteotomy is utilized for correction of pes cavovarus (Figure 4C,D) [17].

### 2.13. Closing-Wedge Osteotomy

This procedure, originally described by Dwyer, is performed in cases of mild pes cavovarus, often in the setting of peroneal tendon pathology, and may be performed alongside peroneal tendon repair [15]. An 8–12 mm wedge of bone is resected via a lateral approach osteotomy, theoretically displacing the weight-bearing axis of the hindfoot more laterally. By reducing stress on the repaired peroneal tendons, closing-wedge osteotomies reduce the chance of tendon re-tear. This is a more challenging technique as the osteotomy and tendon repair are often performed through the same incision.

### 2.14. Rotational Osteotomy

Evans described a lengthening opening-wedge osteotomy of the calcaneal neck to address an overcorrected clubfoot deformity [16]. A wedge-shaped bone allograft is inserted into the osteotomy site, which lengthens the lateral column and rotates the hindfoot and forefoot medially to restore the arch of the foot. This may be performed together with a medial translational osteotomy for correction of severe pes planovalgus. Because the Evans osteotomy accentuates the equinus deformity and leads to varus forefoot alignment, it is commonly performed with a medial cuneiform osteotomy (Cotton osteotomy) to correct forefoot varus. Additionally, a lengthening gastrocnemius/soleus procedure is often performed with the Evans osteotomy to correct ankle equinus (Figure 4E).

Radiographic evaluation of calcaneal osteotomies consists of describing the degree of alignment correction, appropriate healing of the osteotomy and incorporation of graft material if present. Weight-bearing radiographs, including the hindfoot (Saltzman) view, aid the radiologist in describing postoperative alignment [18].

Nonunion or malunion are rare complications associated with calcaneal osteotomies. According to Greenfield et al., this most commonly occurs in patients with systemic comorbidities, including vitamin D deficiency. At-risk patients may receive biologic supplementation in the form of osteoinductive modalities such as bone morphogenetic protein (BMP) at the osteotomy site in hopes of achieving better postoperative outcomes [19].

### 2.15. Medial Cuneiform Opening-Wedge Osteotomy

Medial cuneiform opening-wedge (Cotton) osteotomy is an adjunct procedure used for correction of the fixed forefoot varus component of adult pes planovalgus. The osteotomy is performed at the midpoint of the medial cuneiform from dorsal to plantar, keeping the plantar cortex intact. A 4–6 mm opening is made in the medial cuneiform to accommodate a wedge-shaped autograft or allograft. In recent years, insertion of trabecular titanium wedges has been proposed in lieu of bone grafts (Figure 5) [20]. The osteotomy is usually fixed with Kirschner wires for 4–6 weeks and the patient is kept non-weight-bearing; permanent screw fixation is usually not required.

Postoperative weight-bearing radiographs are utilized to evaluate osteotomy healing and graft incorporation. Meary’s (talus-first metatarsal) and Kite’s (talus-calcaneal) angles are measured on lateral and anteroposterior radiographs to assess restoration of the plantar arch.

## 3. Arthrodeses

### 3.1. Hindfoot

The hindfoot joints include the talonavicular, subtalar, and calcaneocuboid joints. Hindfoot arthrodesis is indicated in symptomatic patients with advanced arthritis or severe hindfoot malalignment (varus or valgus) who have failed conservative management [21,22]. Hindfoot deformity can be painful even in the absence of significant arthritis. In advanced Charcot–Marie–Tooth disease, hindfoot varus can result in symptomatic adult clubfoot. Inflammatory disease such as rheumatoid arthritis can also lead to both deformity and arthritis. Due to the importance of the hindfoot joints in the biomechanics of the foot and ankle, arthrodesis should not be attempted unless other treatment strategies such as tendon transfers, corrective osteotomies, and midfoot arthrodesis have failed.

### 3.2. Subtalar

Subtalar arthrodesis is performed in the setting of severe arthritis of any etiology (primary, post-traumatic, postseptic, or inflammatory) or in other conditions such as talocalcaneal coalition and acquired pes planus deformity related to posterior tibial tendon dysfunction.

Because of the vital role of this joint in foot and gait biomechanics, anatomic alignment of the fusion is essential with a goal of 5–10 degrees of valgus. Open subtalar arthrodesis is generally preferred for more severe deformities and when additional arthrodeses are planned. Fixation is achieved with two large cannulated screws. One of the screws crosses the joint towards the talar neck. The second screw is placed parallel to the sagittal plane, superior to the first screw into the subchondral bone of the talar dome.

Postoperative radiographs are acquired to assess for degree of fusion, alignment, and possible complications, including hardware failure (Figure 6A–D).

### 3.3. Triple Arthrodesis

Triple arthrodesis involves fusion of the talonavicular (TN), talocalcaneal (TC), and calcaneocuboid (CC) joints (Figure 7). The primary goals of a triple arthrodesis are to relieve pain from arthritic, deformed, or unstable joints. Other goals are the correction of deformity and the creation of a stable, balanced plantigrade foot for ambulation. A variety of techniques exist for this procedure and a combination of screws and plates may be used [23]. Complications include nonunion, hardware failure, malalignment, and adjacent joint arthritis. In the late postoperative period, special attention should be directed to hardware integrity and developing arthritis in the unfused joints.

### 3.4. Tibiotalar

The goal of tibiotalar arthrodesis is to solidly fuse the tibia and talus in appropriate alignment, defined as 0–5 degrees of valgus, neutral dorsiflexion, and slight external rotation, in order to obtain pain-free weight-bearing and gait improvement. There is currently no consensus on the best surgical approach for this procedure; hence, a variety of techniques exist. This procedure is performed both open and arthroscopically. The arthroscopic approach is usually selected in patients with minimal-to-no malalignment, whereas the open approach is preferred for patients with moderate-to-severe deformity for better joint visualization and alignment correction. Both internal and external fixation may be used in tibiotalar arthrodesis; successful outcomes have been demonstrated with both methods [24,25].

Various methods of internal fixation have been described, including screws, plates, and retrograde intramedullary nails. Many surgeons prefer to use screws as the primary means of internal fixation because screws are easy to use, have low morbidity, and are less expensive compared to most other methods. However, higher nonunion rates of the ankle joint have been reported with screw fixation, especially in demineralized bone [26]. Plates are advantageous for ankle arthrodesis because they are stiffer constructs than screws and may achieve better union rates; there are also many options available in terms of plate size and shape. However, the extensive dissection needed to place a plate can lead to a higher risk of infection and morbidity [27,28]. A combination of plates and screws may also be used. A recent biomechanical study found that a combination of plates and screws provided significantly greater stiffness than plates or screws alone [28]. Retrograde intramedullary arthrodesis is typically reserved for arthrodesis of both the ankle and subtalar joints.

An osteotomy of the distal fibula is often made proximal to the ankle joint. The resected fibular bone block can be discarded or kept for use as an autologous bone graft later on in the case. The tibiotalar joint is typically fixed using two to three cannulated screws after adequate alignment is obtained. Fusion of lateral malleolus to tibia is then performed using two screws. The fibular bone graft may then be used about the fusion site to facilitate union (Figure 6E).

Radiographs are obtained at regular intervals postoperatively to assess for complications at the fusion site and the adequacy of osseous union. The development of osteoarthritis in adjacent joints should also be monitored.

## 4. Osseous Resection

### 4.1. Accessory Navicular Resection

Though the majority of patients with an accessory navicular are asymptomatic, a small fraction of patients may develop pain, which is thought to be related to traction from the posterior tibialis tendon (PTT). A type II accessory navicular (triangular or semicircular ossicle attached to the navicular by a fibrocartilage or hyaline-cartilage synchondrosis) is most likely to become symptomatic [29,30]. Patients are most often managed conservatively, but for refractory cases the accessory navicular may be surgically resected. Surgical techniques can be described with regard to their degree of manipulation of the PTT. In simple excision, the dorsal PTT is lifted away from the ossicle and the plantar PTT is not disturbed. Any dissected PTT is reattached to the navicular proper. In the Kidner procedure the ossicle is resected, the PTT is detached and advanced through a tunnel in the medial navicular and sutured to itself and the navicular periosteum [31,32].

Preoperative radiographs may demonstrate degenerative changes across the synchondrosis associated with a type II accessory navicular. Magnetic resonance imaging (MRI) is superior in detecting bone-marrow edema in the ossicle and/or navicular proper as well as posterior tibial tendinosis. Postoperative radiographs are acquired to confirm complete resection of the ossicle. Postoperative MRI may be obtained as indicated to evaluate the integrity of the PTT (Figure 8).

### 4.2. Cheilectomy

Cheilectomy (derived from the Greek word, *cheil*, meaning tongue) involves the resection of dorsal osteophytes from the first metatarsal head. Sometimes as much as one-third of the dorsal-metatarsal head bone stock is resected [33]. Typically, cheilectomy is performed for mild-to-moderate first MTP arthritis and results in pain relief and improved range of motion. For severe first MTP arthritis, arthrodesis is preferred. Postoperatively, lateral radiographs are helpful in detecting subtotal resection or osteophyte recurrence.

## 5. Articular Implants

### 5.1. Polyvinyl Alcohol Hydrogel Hemiarthroplasty/Synthetic Cartilage Implant (SCI)

The goal of polyvinyl alcohol (PVA) hydrogel hemiarthroplasty, commonly known by its trade name Cartiva, is to relieve pain and improve range of motion in patients with first MTP osteoarthritis and/or moderate-to-severe hallux rigidus [33,34]. It can be performed in patients with mild hallux valgus deformities. The implant is made of polyvinyl alcohol, a synthetic polymer with biochemical properties similar to cartilage [35]. A precisely measured cylindrical hole is drilled through the central cartilage and subchondral bone of the metatarsal head and the implant is pressed into place without screws or cement. Cheilectomy is often performed concurrently. The phalangeal side of the joint remains unaltered (Figure 9A).

Postoperative radiographs are obtained at regular intervals to assess for complications, including implant displacement, loosening, subsidence, and recurrent osteophyte formation. Radiographically, the implant appears as a geographic rectangular lucency in the first-metatarsal head. Cross-sectional imaging better demonstrates the cylindrical shape of the implant.

### 5.2. Silastic Implant

Arthroplasty utilizing high-performance silicone implants, termed silastic implants, was originally performed in joints of the hand. In the foot, a single-stem silastic implant for a first MTP hemi-arthroplasty was first used in 1968. The procedure has undergone multiple iterations since that time, with current techniques utilizing double-stem, hinged implants [36]. Indications include severe inflammatory arthritis or osteoarthritis, with or without hallux limitus or hallux ridigus [36,37,38,39]. Some authors advocate for the use of this technique in older, less active patients, as this population demonstrates the best clinical outcomes with fewer complications [38]. Prerequisites for silastic-implant placement include intact collateral ligaments and sufficient bone stock. After resection of the articular surface, the medullary cavity is opened, taking care to preserve the collateral ligaments. A trial prosthesis is placed, and the joint is manipulated under fluoroscopy to assess for joint subluxation. The final prosthesis is then inserted with a press-fit technique. The implant acts as a spacer and is not designed to function as a joint replacement. Specific attention must be given to the rebalancing of the soft-tissue structures in an effort to restore the balance of the MTP [36].

Radiographically, silastic implants have increased density relative to the adjacent bone (Figure 9B). The articular surfaces are flat and should closely oppose one another. The component stems are triangular in shape and should be flush with the native bone. Postoperative radiographs should be scrutinized for implant fracture, periprosthetic fracture, and signs of osteolysis or implant loosening [40]. Importantly, the presence of periprosthetic cystic changes and osteolysis may be seen in a large proportion of asymptomatic patients.

### 5.3. Subtalar Arthroereisis

Patients with flexible flatfoot deformity (hindfoot valgus, talar plantar flexion, and longitudinal arch collapse) who fail conservative management may benefit from surgical intervention [41]. Arthroereisis is a surgical procedure where a bioabsorbable or titanium implant is inserted into the sinus tarsi, expanding the subtalar joint and biomechanically restricting flatfoot deformity [42,43,44]. The name is derived from the Greek root -ereisis, translated as the action of supporting or lifting up.

Preoperative weight-bearing radiographs demonstrate the degree of flatfoot deformity and the morphology of the subtalar joint, and allow for related presurgical measurements. Intraoperatively, the size of the implant is determined based on the range of motion of the subtalar joint. Intraoperative fluoroscopic or radiographic images are used to assess implant alignment and ensure adequate correction of the deformity. Postoperative radiographs demonstrate the arthroereisis as a radiodense cylindrical implant located between the anterior and posterior subtalar facets near the angle of Gissane (Figure 10). On the lateral radiograph, the tip of the implant should be within the subtalar joint with its long axis parallel to the joint space. On anteroposterior radiographs of the foot, the implant should project over the middle third of the talus. Its lateral margin should align with or be slightly medial to the lateral margin of the calcaneus. Postsurgical weight-bearing radiographs demonstrate the degree of operative correction.

Complications such as implant loosening, dislocation, lateral extrusion, and overcorrection of the deformity may be appreciated radiographically (Figure 11) [43]. Subtle implant migration, fractures, and peri-hardware lucency may be better evaluated with CT, whereas MRI may demonstrate postoperative soft-tissue abnormalities or bone-marrow edema [43]. Patients may complain of postoperative pain from sinus tarsi syndrome or accelerated subtalar osteoarthritis, which may be severe enough to require implant removal (Figure 12) [45].

### 5.4. Total Ankle Arthroplasty

Total ankle arthroplasty was developed as an alternative to arthrodesis in patients with end-stage tibiotalar arthrodesis who failed conservative management, with the goals of decreasing pain and restoring ankle alignment while maintaining range of motion. Several generations of these implants exist, with the second and third generations being the most commonly used today. The implants are divided into mobile-bearing and fixed-bearing models based on whether or not the polyethylene spacer is mobile or fixed to the tibial component [46,47].

As with all implants, postoperative evaluation is focused on potential complications, not dissimilar to other arthroplasties, and includes periprosthetic lucency/fracture, osteolysis, and subsidence (Figure 13) [48].

## Figures and Tables

**Figure 1 diagnostics-12-01090-f001:**
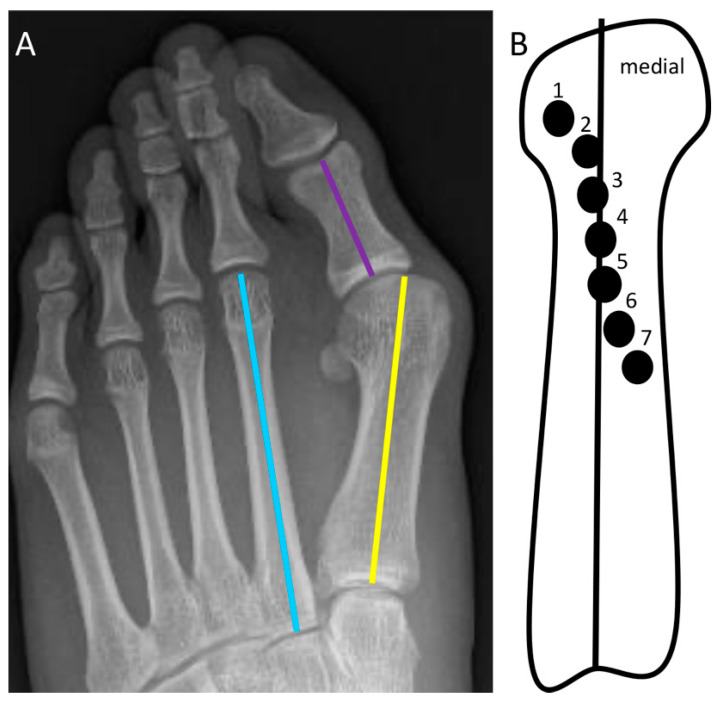
(**A**) Anteroposterior (AP) radiograph of the foot demonstrating the longitudinal axis of the first metatarsal (yellow line), first proximal phalanx (purple line), and second metatarsal (blue line). IMA is the angle between the first and second metatarsal longitudinal axes. HAA is the angle between the first metatarsal and first proximal phalangeal longitudinal axes. (**B**) Schematic of MSP, typically measured on a 7-point scale that increases with progressive medial angulation of the first metatarsal.

**Figure 2 diagnostics-12-01090-f002:**
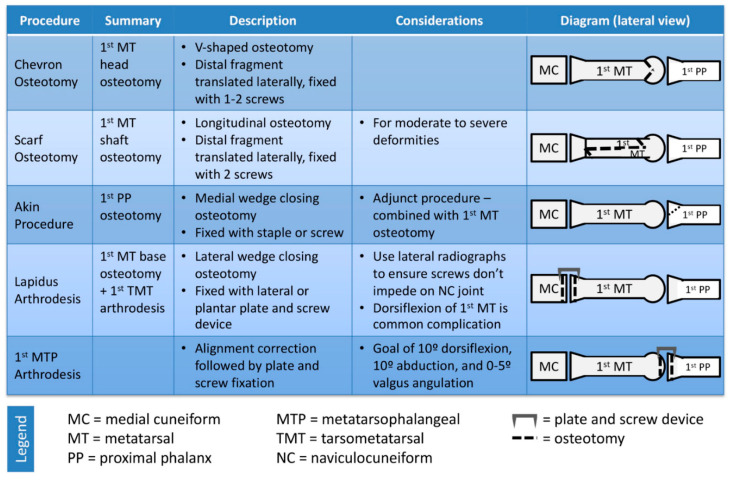
Surgical options for correction of hallux valgus.

**Figure 3 diagnostics-12-01090-f003:**
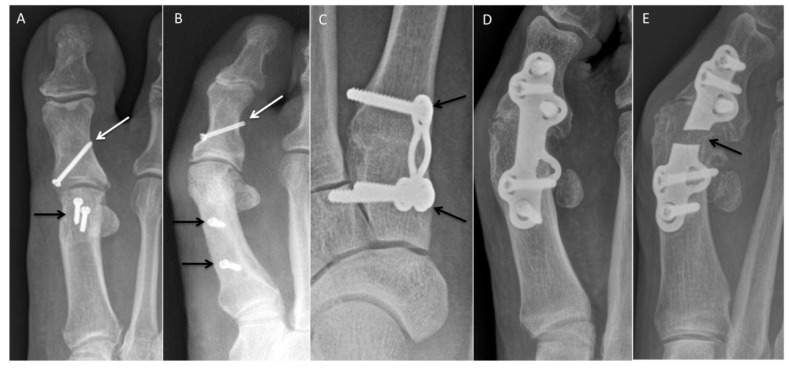
Hallux valgus correction. (**A**) Postoperative AP radiograph demonstrating Chevron (black arrow) and adjunct Akin (white arrow) correction of hallux valgus deformity with first metatarsal head and proximal phalangeal base osteotomies. Two first-metatarsal head screws and one screw in the base of the first proximal phalanx transfix the osteotomy sites. (**B**) Postoperative AP radiograph demonstrating Scarf (black arrows) and adjunct Akin (white arrow) correction of hallux valgus deformity with first-metatarsal shaft and proximal phalangeal base osteotomies. (**C**) Postoperative changes related to Lapidus procedure with first-metatarsal base osteotomy and first TMT arthrodesis (black arrows). (**D**) First MTP arthrodesis for correction of hallux valgus. (**E**) Postoperative complication of first MTP arthrodesis with fractured plate (black arrow).

**Figure 4 diagnostics-12-01090-f004:**
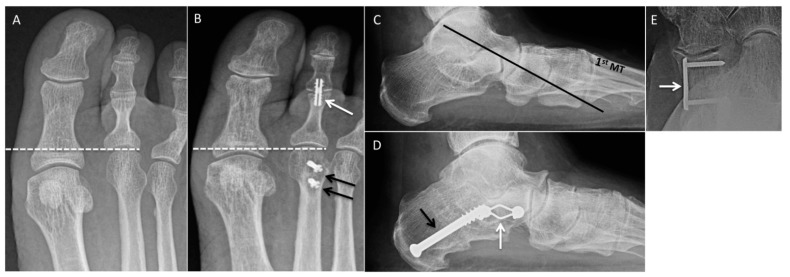
Osteotomies. (**A**) Preoperative AP radiograph demonstrating long-second metatarsal in a patient with plantar pain. (**B**) Postoperative radiograph showing findings of Weil osteotomy with screw fixation (black arrows) with decreased length of the second metatarsal (white dashed lines). Additionally, the patient underwent second PIP arthrodesis (white arrow). (**C**) Preoperative lateral radiograph demonstrating altered tarsometatarsal axis (black line) passing plantar to the first metatarsal. Postoperative lateral (**D**) and AP (**E**) radiographs demonstrating healed medializing calcaneal osteotomy (black arrow) and Evans lengthening osteotomy (white arrows).

**Figure 5 diagnostics-12-01090-f005:**
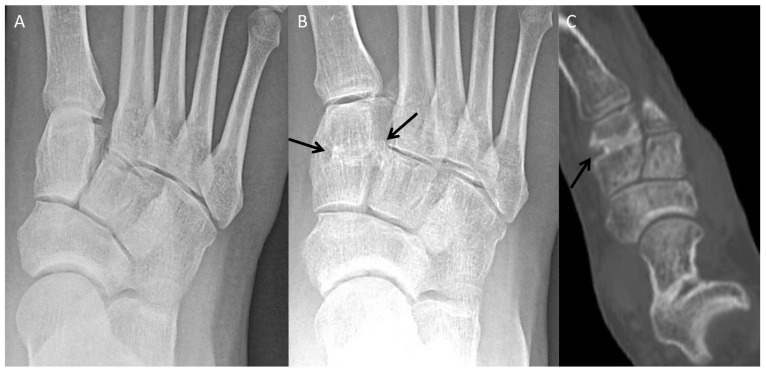
(**A**) Preoperative and (**B**) postoperative AP radiographs, and (**C**) axial CT of the foot demonstrating changes of Cotton osteotomy at the waist of the medial cuneiform with interposition of bone graft material (black arrows).

**Figure 6 diagnostics-12-01090-f006:**
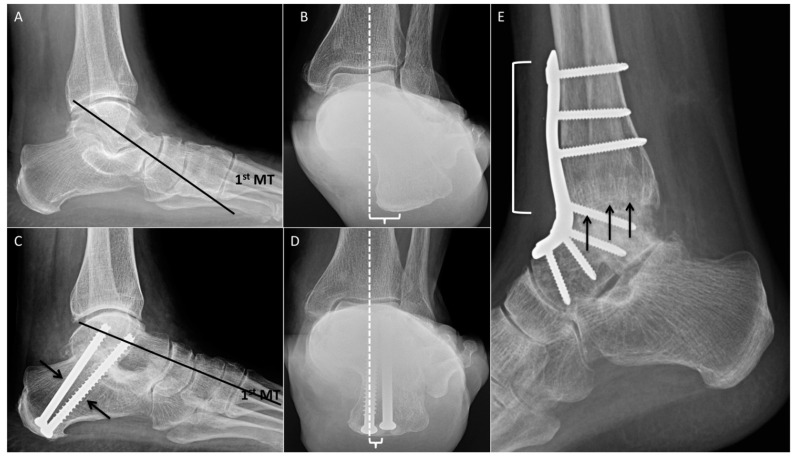
Arthrodeses. (**A**,**B**) Preoperative weight-bearing radiographs of the hindfoot. (**A**) Weight-bearing lateral radiograph demonstrating the longitudinal axis of the talus (black line) passing plantar to the first metatarsal (MT), consistent with severe pes planus. There is also mild subtalar osteoarthritis. (**B**) Saltzman view of the hindfoot demonstrating the weight-bearing axis of the tibia (white dashed line) passing far medial to the plantar calcaneus, compatible with hindfoot valgus. (**C**,**D**) Postoperative weight-bearing radiographs of the hindfoot following subtalar arthrodesis. (**C**) Weight-bearing lateral radiograph demonstrates improved alignment, with the longitudinal axis of the talus (black line) passing through the first MT. Note the talocalcaneal fixation screws (black arrows). (**D**) Saltzman view of the hindfoot demonstrating resolved hindfoot valgus, with the weight-bearing axis of the tibia (white dashed line) passing near the plantar calcaneus. (**E**) Postoperative lateral radiograph demonstrating tibiotalar arthrodesis with plate and multiple screws (white bracket). There is partial ankylosis (black arrows).

**Figure 7 diagnostics-12-01090-f007:**
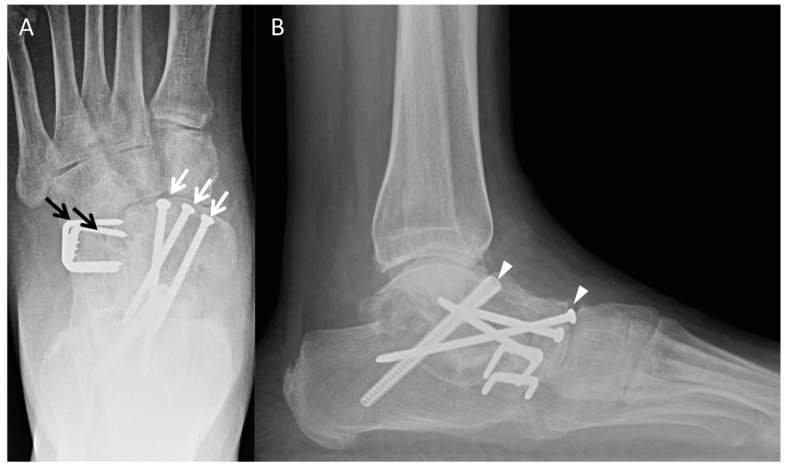
(**A**) AP and (**B**) lateral radiographs of the foot demonstrate hindfoot triple arthrodesis with hardware spanning the subtalar (arrow heads), talonavicular (white arrows) and calcaneocuboid joints (black arrows). The hardware is intact and there is mature ankylosis with decreased conspicuity of the joint space.

**Figure 8 diagnostics-12-01090-f008:**
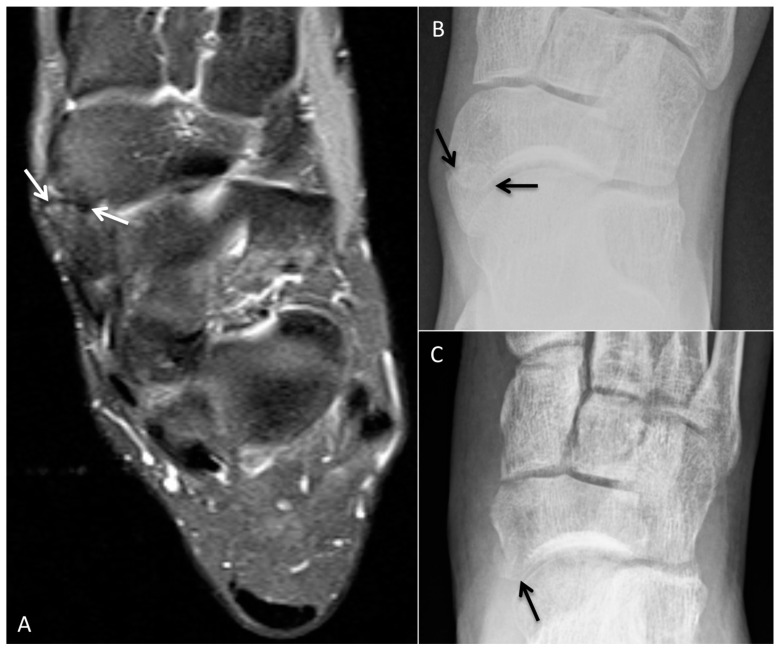
Pre- and postoperative images in a patient with a symptomatic accessory navicular. (**A**) Preoperative axial PD fat-suppressed MR image with mild edema/degenerative changes at the accessory navicular synchondrosis (arrow). (**B**) Preoperative radiograph with type II accessory navicular (arrows). (**C**) Postoperative changes from accessory navicular resection without complication (arrow).

**Figure 9 diagnostics-12-01090-f009:**
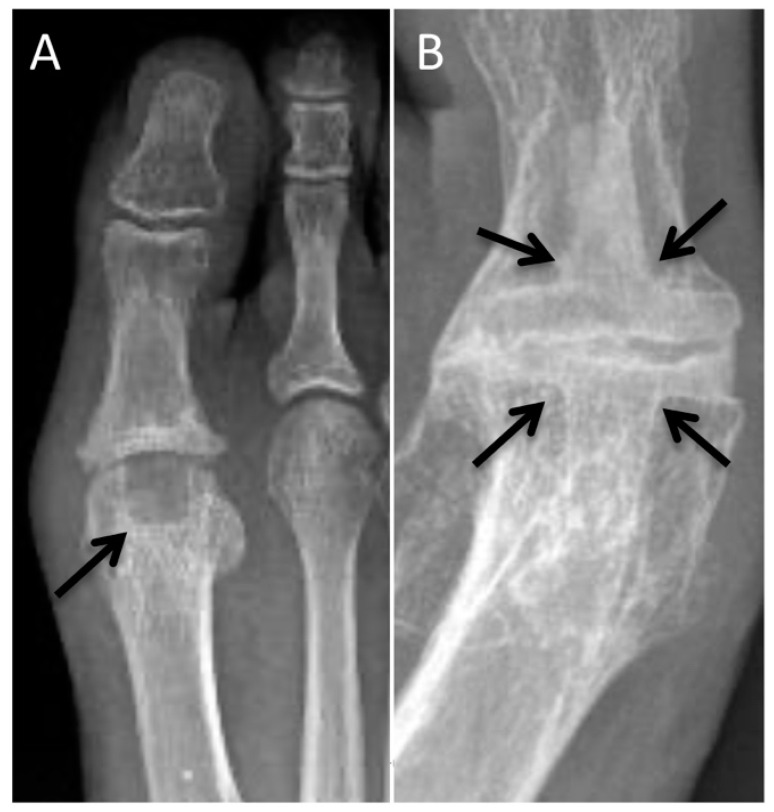
Implants. (**A**) AP radiograph demonstrating postoperative changes related to Cartiva implant (arrow) with geographic rectangular lucency in the first metatarsal head. (**B**) AP radiograph showing silastic implant (arrows) of the first MTP joint. Note flat articular surfaces and triangular shape of stems.

**Figure 10 diagnostics-12-01090-f010:**
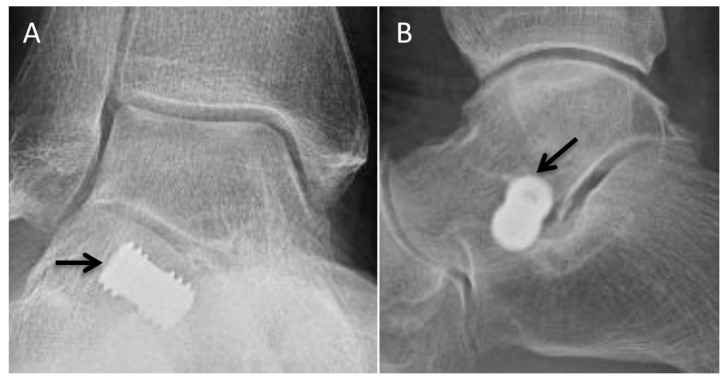
(**A**) AP and (**B**) lateral radiographs demonstrating the arthroereisis screw expanding the subtalar joint (black arrows) positioned between the anterior and posterior subtalar articulations.

**Figure 11 diagnostics-12-01090-f011:**
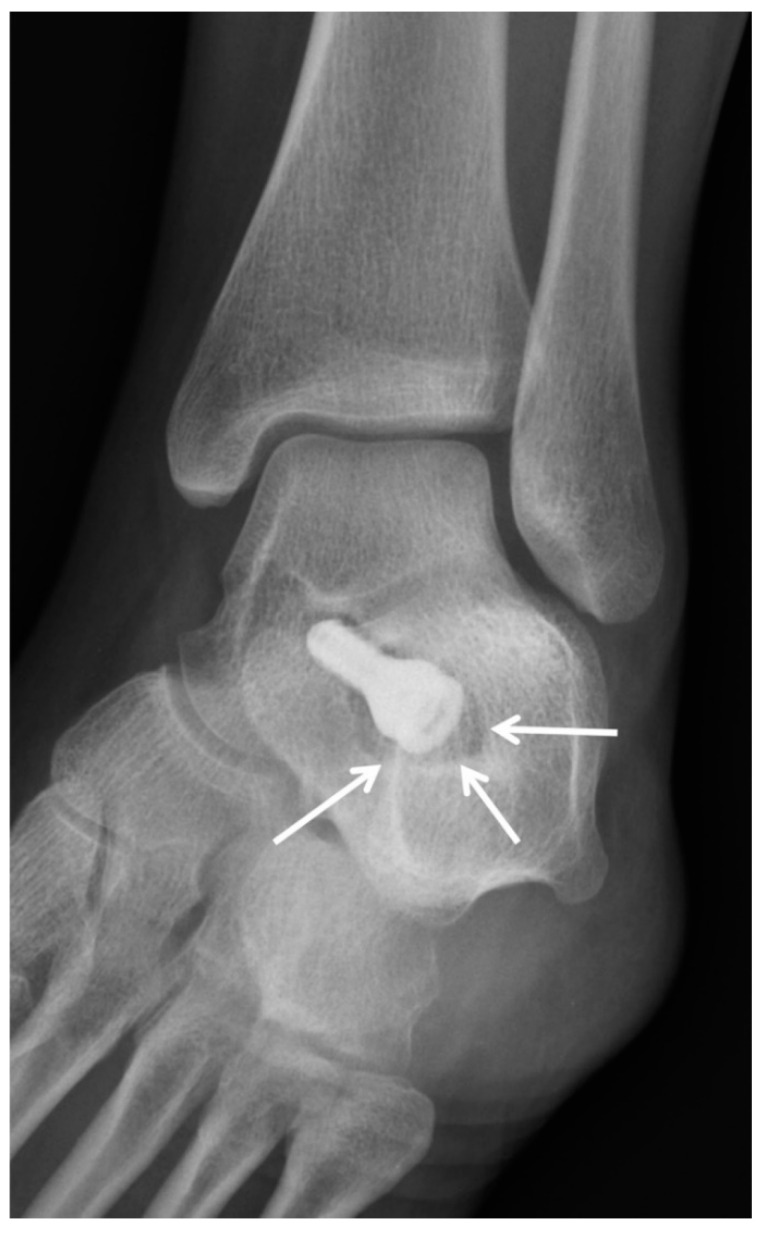
Prior history of subtalar arthroereisis placement. Mortise radiograph of the ankle demonstrates increased lucency surrounding the hardware concerning for loosening (white arrows).

**Figure 12 diagnostics-12-01090-f012:**
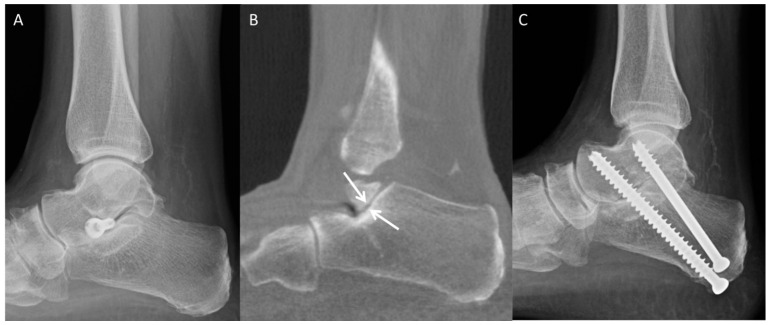
Prior left arthroereisis placement (**A**) developed hindfoot pain. Weight-bearing CT (**B**) demonstrates accelerated posterior subtalar facet degenerative changes (white arrows). Patient underwent implant removal and subsequent subtalar arthrodesis (**C**) with resolution of pain.

**Figure 13 diagnostics-12-01090-f013:**
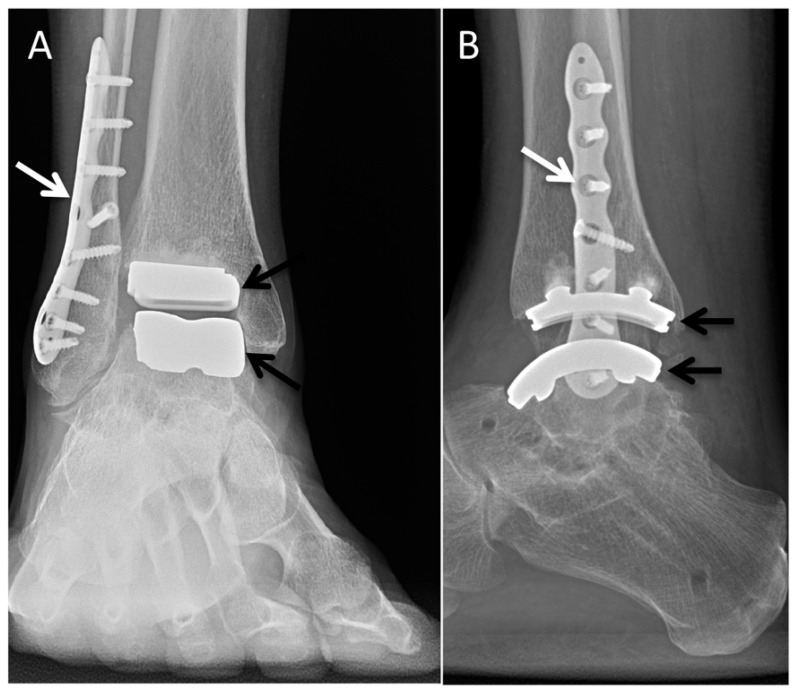
Mobile-bearing total ankle arthroplasty. AP (**A**) and lateral (**B**) radiographs demonstrating well-seated tibial and talar components (black arrows) without abnormal lucency or fracture. Fibular osteotomy (white arrows).

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
