# Peer review of "Postoperative Findings of Common Foot and Ankle Surgeries: An Imaging Review"

_diagnostics, 2022, doi:10.3390/diagnostics12051090_

Round 1

Reviewer 1 Report

The manuscript reviews common procedures used for ankle and foot corrective surgeries along with their radiographic appearances. The manuscript is well-written and I enjoyed reading it. Overall, no major comments as the scientific background are solid. The paper, however, may benefit from proofreading for style errors, e.g: font changes, extra spaces, and misplaced punctuations.

One minor comment: Please check if the diagram for Akin procedure in Table 1 is correct.

Author Response

Akin procedure diagram in Table 1 corrected.

Additional proofreading performed. 

Reviewer 2 Report

very interesting article.
if possible add images on Triple Arthrodesis and  add other types of subtalar arthroeresis.
I would also increase the traumatic part. or delete the paragraph and avoid the traumatic part

Author Response

Additional images of the triple arthrodesis and subtler arthroereisis were added.

Trauma section was removed.